# Estimating the costs of blindness and moderate to severe visual impairment among people with diabetes in India

Stuart Redding ,[1] Robert Anderson ,[1] Rajiv Raman ,[2]
Sobha Sivaprasad ,[3] Raphael Wittenberg ,[1] The ORNATE India Project Group

¹Centre for Health Service Economics & Organisation, Nuffield Department of Primary Care Health Sciences, University of Oxford, Oxford, UK
²Shri Bhagwan Mahavir Vitreoretinal Services, Sankara Nethralaya, Chennai, India
³Clinical Research Facility, Moorfields Eye Hospital, London, UK

**Correspondence to**
Dr Stuart Redding;
stuart.redding@phc.ox.ac.uk

## ABSTRACT

**Objectives** This study provides an estimate of the annual cost of blindness and moderate to severe visual impairment (MSVI) among people with diabetes aged 40 years and above in India in the year 2019.

**Design** A cost of illness study.

**Setting** India.

**Participants** People with diabetes aged 40 years and above in India in the year 2019.

**Primary and secondary outcome measures** Estimates are provided for the total costs of screening for most common vision-threatening eye conditions, treatment of these conditions, economic activity lost by these people and their family carers whose ability to work is affected, and loss of quality of life experienced by people with diabetes and blindness or MSVI.

**Results** It is estimated that for people with diabetes aged 40 years or above, annual screening followed by eye examination where required would cost around 42.3 billion Indian rupees (INR) (4230 crores) per year; treating sight problems around 2.87 billion INR (287 crores) per year if 20% of those needing treatment receive it; and lost economic activity around 472 billion INR (47 200 crores). Moreover, 2.86 million (0.286 crores) quality-adjusted life years (QALYs) are lost annually due to blindness and MSVI. The estimate of lost production is highly sensitive to the proportion of people with MSVI able to work and how their output compares with that of a person with no visual impairment.

**Conclusions** This is the first study to estimate the cost of blindness and MSVI for people aged 40 years and over with diabetes in India. The annual cost to the Indian economy is substantial. This cost will be expected to fall if a successful screening and treatment plan is introduced in India. Further work is suggested using more robust data, when available, to estimate the loss of productivity and loss of QALYs, as this would be worthwhile.

## STRENGTHS AND LIMITATIONS OF THIS STUDY

⇒ This is the first study to estimate the cost of blindness and moderate to severe visual impairment for people aged 40 years and over with diabetes in India.
⇒ A range of data are employed including recent data on costs of screening for diabetic retinopathy in Kerala, but it is not possible to be certain that the costs of screening would be similar across other states in India.
⇒ Care has been taken to make realistic assumptions, in particular that it would not be realistic for the full backlog of people needing treatment to all be treated in just one year.
⇒ Two alternative sets of assumptions are made regarding lost productivity; collection of data would be valuable to enable a more robust estimate to be made of lost productivity.
⇒ The data on the quality of life of people with eye conditions are drawn from a recent study in India, but the sample with severe sight problems is fairly small.

## INTRODUCTION

Blindness, defined by the WHO as Snellen visual acuity of 3/60 or worse in the better eye,[1] is a debilitating condition which, in addition to the disabling consequences for the person affected, generates substantial care and treatment costs for society. While some people who suffer from vision loss are able to work, their productivity is likely to be less than that of a person with better vision and some may drop out of the labour force entirely. Therefore, blindness or sight loss reduces the potential output of the economy by decreasing the output of both those who are directly affected and those who have to stop or reduce work in order to provide unpaid care for people who are blind.

Significant work has been undertaken to measure and reduce the prevalence of blindness in India over the last 20 years. Much of this work has been influenced by or undertaken directly as part of the Vision 2020 initiative implemented by the WHO, which aimed to eliminate avoidable blindness by 2020,[2] and the WHO Global Action Plan for Universal Eye Health 2014–2019, which set a target of a reduction in the prevalence of visual impairment by 25% by 2019 from the baseline level of 2010. Actions taken have facilitated a 47.1% reduction in the prevalence of blindness, 52.6% in moderate to

severe visual impairment (MSVI) and 51.9% in visual impairment during this time period, mainly by reducing the cataract burden and by correcting refractive errors.[3]

One of the main preventable causes of blindness is diabetic retinopathy (DR). This is a microvascular complication of diabetes that can cause diabetic macular oedema or proliferative DR, both of which can result in vision impairment and progress to irreversible sight loss. Together they are termed sight-threatening DR (STDR). Among individuals with diabetes, the global prevalence of DR was 22.3% in 2020.[4] As life expectancy continues to increase, the numbers of people with diabetes and complications such as DR are expected to rise unless people adopt healthier lifestyles or preventive interventions become available.[5] In addition, people with diabetes are also affected by other common causes of blindness such as cataract and glaucoma, and so rates of blindness in people with diabetes exceed those of people without diabetes.

This study reports the findings of a cost of illness study in which an attempt to quantify an annual value of the total economic impact of blindness and treatment of sight-threatening conditions among people with diabetes aged 40 years and over in India is made. Cost of illness studies identify and measure direct, indirect and intangible costs associated with a disease and show the total burden of that disease on society (see, for example, Larg and Moss[6] for a summary of the approaches used and their limitations). This cost of illness study estimates the total costs of screening people aged 40 years and over with diabetes, healthcare for those found to have eye conditions and lost productivity by those who are blind or experience MSVI and by family and friends providing unpaid care for them. Data on the prevalence of blindness and MSVI among people with diabetes; service utilisation by those with eye conditions and associated cost data from multiple sources; and economic activity lost due to blindness or MSVI by people with diabetes and by those who stop or reduce work to care for them are used. An estimate of the loss in quality of life (QoL) experienced by people with diabetes who have blindness or MSVI is also provided.

Monitoring and screening patients with diabetes for DR also identify other eye conditions that cause visual impairment that require treatment. Therefore, the costs of treatment of cataract and glaucoma as well as of DR among people with diabetes presenting with MSVI or blindness are included.

This cost of illness study is part of the ORNATE India research project,[7] which aims to evaluate cost-effective measures for screening for diabetes and its complications and to examine the potential impact of a reduction in the prevalence of blindness on the Indian economy. The project includes an evaluation of a pilot screening programme for DR in Kerala, the Nayanamritham Project,[8 9] and an evaluation of a community screening programme for diabetes and its complications in 20 areas of India covering all the six regions, the SMART India Project.[10]

The direct medical cost of DR alone has been explored in countries such as the USA,[11] Hungary[12] and Singapore.[13] There has also been cost of illness analysis of type 2 diabetes and its complications in lower/middle-income countries such as Iran.[14] The economic impact of DR alone has been measured in Germany[15] and the USA,[16] among other countries, but, to the best of our knowledge, this is the first attempt to adopt a holistic and pragmatic approach for India incorporating all common causes of blindness and MSVI that can affect people with diabetes.

Studies to measure the economic impact of all-cause blindness in India have been limited. The most recent complete estimate of the economic cost of blindness in India[17] of which we are aware was produced more than 20 years ago. That paper produced an estimate of the burden of blindness in India of 159 billion Indian rupees (INR) for 1997, and a cumulative lifetime loss of 2787 billion INR. It also estimated that treating all cases of blindness due to cataract would cost 5.3 billion INR but did not evaluate the impact of STDR in people with diabetes.

The estimate provided here relates to the annual cost of retinal screening for people aged 40 years and over who have diabetes in 2020, identifying and treating all common causes of preventable blindness and MSVI and lost productivity by people with diabetes who are blind or have MSVI and their unpaid carers. This is known as the prevalence approach to cost of illness studies. The incidence approach, which would provide an estimate of the lifetime costs for a cohort of people with diabetes-related blindness, is not adopted.

The costs of screening and treatment (direct costs) and economic costs of lost productivity (indirect costs) are related to different cohorts. The direct costs are related to people who are referred to hospitals for treatment after being identified during the screening programme as having eye conditions and the indirect costs are related to people who are blind or have MSVI. It is important to understand the overall annual costs incurred by the Indian economy covering both those relating to screening and treatment, including those incurred by patients and those relating to lost productivity that would be avoided through screening and treatment. Understanding the overall economic consequences of blindness is crucial to ensure that the case for devoting resources to the prevention, treatment and research on eye disease is understood and justified.

## RESEARCH METHODS
### Overall approach
Societal costs of blindness and MSVI among people with diabetes aged 40 years and over in India in 2019, including costs of screening and healthcare and of lost economic output for blind people with diabetes, people with MSVI and diabetes and their carers, are estimated. Results for these three categories of costs are provided separately. All reported estimates are annual costs for India, in INR at 2019 prices (although some of the prevalence data are

related to 2020). Cost estimates represent a snapshot for the year and not lifetime costs. Discounting was unnecessary since all costs refer to a 1-year period.

## Data sources
Estimates were derived using data from multiple sources as discussed below.

### Number of people affected
First, the number of people aged 40 years and over who have diabetes and people who are 50 years and over who have diabetes and blindness or MSVI due to existing eye conditions is defined.

Age-specific prevalence data for diabetes are drawn from the Global Burden of Disease (GBD) Study[18] and combined with 2020 population data from PopulationPyramid.net for 2020 (www.PopulationPyramid.net[19]) to provide an estimated aggregate number of people in India aged 40 years and over with diabetes in the year 2020. A recent study by the GBD[20] showed prevalence rates for a number of health conditions for South Asia. As most of the population in South Asia is from India, it is appropriate to use these values to estimate the number of cases in India. The age-standardised prevalence of blindness in people aged 50 years or above in South Asia is 29.8/1000.

The GBD 2020 prevalence rates per 1000 population aged 50 years and over for common eye conditions among people with blindness and people with MSVI are set out in table 1. Also shown are estimates of the number of people (in thousands) aged 50 years and over with diabetes and each of the eye conditions. These estimates assume that, with the exception of DR, which can only affect people with diabetes, the prevalence of each eye condition is the same for people with diabetes as for the wider population. The prevalence rates in the table, which relate to total population aged 50 years and over, imply DR rates of 3.07 among people with diabetes who are blind and 7.86 among people with diabetes who have MSVI. Summing

up the eye conditions implies that there are around 1 335 000 blind people aged 50 years and over with diabetes and 9 096 000 people aged 50 years and over with MSVI and diabetes.

### Costing care
Blindness can be treated in some conditions such as cataract but not in others including advanced age-related macular degeneration and DR. However, there are treatments available to slow the progression and/or prevent blindness and MSVI due to most of these common conditions. Measuring the aggregate costs of these treatments in India is attempted. The costs of care are based on evidence from previous literature, expert advice and calculations derived from a pilot scheme run to screen for DR among people with diabetes in Kerala,[10] which was conducted as part of the ORNATE India research project described in the Introduction section.

Data from the Nayanamritham Study in Kerala suggest a unit cost of initial screening of approximately 400 INR, if all people with diabetes in the non-communicable disease register are screened. This estimate includes 85 INR staff costs, 25 INR cost of administrative support, 200 INR annuitised cost of the cameras and 105 annuitised cost of staff training in the Kerala pilot. Information on staff salaries, staff time per person screened, costs of cameras and costs of staff training was provided by the Kerala team. Further details are reported elsewhere (Wittenberg et al[21]).

Furthermore, in this study, 31% of screened patients were referred to hospital services. These may be due to cataract, glaucoma, STDR or ungradable retinal images. The estimated unit cost of these referrals was 1150 INR comprising 200 INR staff costs of the eye examination and 950 INR travel and lost income costs to the patient and their attendant.

The cost of treating those who were found through the screening programme to have sight-threatening

**Table 1** Numbers of people with diabetes in India suffering from blindness or MSVI by eye conditions

| | Number of people (50+) | Aged 50+ crude prevalence (per 1000 people) | | |
|---|---|---|---|---|
| With diabetes in India | 41 194 000 | 153.86 | | |
| | Number of people with diabetes and blindness (50+) | Aged 50+ crude prevalence (per 1000 people) | Number of people with diabetes and MSVI (50+) | Aged 50+ crude prevalence (per 1000 people) |
| All causes | 1 335 000 | 29.80 | 9 096 000 | 33.98 |
| Due to cataract | 758 000 | 18.40 | 3 489 200 | 84.70 |
| Due to glaucoma | 74 000 | 1.79 | 121 900 | 2.96 |
| Due to age-related macular degeneration | 38 000 | 0.92 | 156 100 | 3.79 |
| Due to STDR | 126 000 | 0.47 | 324 000 | 1.21 |
| Due to uncorrected refractive error | 125 200 | 3.04 | 4 119 500 | 100.00 |
| Due to residual causes | 214 000 | 5.20 | 885 700 | 21.50 |

MSVI, moderate to severe visual impairment; STDR, sight-threatening diabetic retinopathy.

**Table 2** Unit cost of treatment (INR)

| Item | Healthcare | Travel | Total cost | Source |
|---|---|---|---|---|
| Screening | 400 | 0 | 400 | Kerala pilot |
| Eye examination | 200 | 950 | 1150 | Kerala pilot |
| Glasses | 1000 | | 1000 | Expert advice |
| Eye drops | 500 | | 500 | Expert advice |
| Cataract surgery | 15 000 | 950 | 15 950 | Expert advice |
| STDR treatment | 12 000 | 2850 | 14 850 | Expert advice |
| Glaucoma surgery | 5000 | 950 | 5950 | John and Parikh[22] |
| Glaucoma average | 1625 | 238 | 1862.5 | Expert advice |
| Age-related macular degeneration | 12 000 | 2850 | 14 850 | Expert advice |

INR, Indian rupee; STDR, sight-threatening diabetic retinopathy.

conditions is also considered. The unit costs, including costs to the patient and their family, are estimated as an average across private providers and health insurance coverage. These include 15 000 INR for cataract surgery, 5000 INR for glaucoma surgery[22] and 12 000 INR for DR treatment (laser or intravitreal injections).[23] The unit costs and sources of our estimates for these interventions are presented in table 2 below. The expert advice was provided by senior clinicians from the SMART India research project.

Cost of blindness estimates for high-income countries take into account the cost of low vision rehabilitation, community care and residential care for people suffering from blindness. However, since these services are in their infancy in India and most patients rely on family carers, costs are not included for them. Costs for hip replacement or depression, which are also included in the costs of blindness in high-income countries, have been omitted from the calculations presented here.

### Costing economic losses

The indirect economic costs of blindness and MSVI accrue from loss of output due to reduced productivity of blind or visually impaired people compared with people with full vision, plus the loss of output by carers who leave employment or reduce their working hours to look after family members or friends who are blind.

Determining an estimate for the aggregate loss of output in India requires use of data relating to hours of work and productivity lost by blind and visually impaired people who could be economically active if they had full sight and hours of work lost by carers who have to cease or reduce work in order to care for blind friends or family members. Some of these data are available and others need to be estimated. Where data are not available, the same assumptions as those adopted by Eckert et al[24] are used. In their study, they assume that no blind people are able to work, and that for each blind person, one potentially active worker forgoes 10% of their productive time to care for the blind person. Furthermore, they assume that all people with MSVI are able to work, but only

produce 70% of what a person with full sight produces. They assume that for each person with MSVI, one potentially active worker forgoes 5% of their productive time to care for the person with MSVI.

Furthermore, people with sight loss over 70 years are assumed to be economically inactive regardless of their vision. For the purpose of our cost estimate, full-time equivalent output per person is valued at the Indian average Gross National Income (GNI) per capita. This was 147 524 INR in 2019.[25] With these assumptions, economic activity lost (EAL in the equation below) can be calculated as:

EAL (blind)=(number of blind people under 70+total number of blind people×10%)×(GNP (Gross National Product) per capita)

EAL (MSVI)=(number of people with MSVI under 70×30%+total number of people with MSVI×5%)×(GNP per capita)

As a sensitivity analysis, an estimate is presented for blind people based on assumptions used by Shamanna et al[17] in their study of all-cause blindness in India. The assumptions taken from Shamanna et al are as follows:
a. 80% of blind people are economically inactive.
b. The average blind person who is economically active produces 25% of the output achieved by a worker with full sight.
c. For each blind person, one economically active worker forgoes 10% of their productive time to care for the blind person.

They do not provide estimates for people with MSVI.

### Calculating quality-adjusted life year losses

Ill-health can affect health-related QoL in many ways. The standard metric for measuring health-related QoL in cost-utility analyses is the quality-adjusted life year (QALY). One QALY is equal to 1 life year in full health; any ill-health that reduces ability to perform day-to-day functions is assumed to reduce QoL, such that 1 life year in less than full health is valued at less than one QALY. The standard QALY metric increasingly used now is the EQ-5D-5L, replacing EQ-5D-3L. Each respondent

indicates on a 5-point scale their ability to function in five specific dimensions (mobility, self-care, usual activities, pain/discomfort and anxiety/depression). Each EQ-5D-5L health state can be assigned a value on a scale or tariff where 1=full health and 0=dead. Since there is currently no value set available for India, the Chinese value set[26] is used in this study.

In order to quantify the QALY losses associated with blindness and with MSVI, individual-level data on these five dimensions combined with data detailing the state of their sight are used. These data are taken from the SMART Study, which was part of the overall ORNATE India Study as outlined above. The SMART Study collected information on EQ-5D-5L health state and on vision using the following self-reported categories:

1. I have no problems seeing.
2. I have slight problems seeing.
3. I have some problems seeing.
4. I have severe problems seeing.
5. I am unable to see.

For the purpose of our analysis, blindness is defined as responses in the categories 'I have severe problems seeing' or 'I am unable to see' and MSVI when the respondent said 'I have some problems seeing'.

To estimate the loss of QALYs due to blindness and MSVI, the response category 'I have no problems seeing' is treated as the counterfactual against which to estimate the loss. The Chinese value set[26] was applied to the EQ-5D data for each person to get a QoL value for each person. Since the aim is to derive the average reduced QoL due to poor sight, it is necessary to establish the average QoL those with poor sight would have had if they had good sight. Since their average age is greater than those with good sight, and QoL level falls with age, the age distribution of those with poor sight is applied to the age-specific QoL values in those with good sight to derive the appropriate comparator—the average QALY if those with poor sight had good sight. Further details of the methods employed in this analysis are presented in the online supplemental material.

### Sensitivity analyses

Sensitivity analysis was performed on three aspects of the analysis.

The costs are estimated based on 10% higher (lower) cost per person for DR screening and eye examination, and also if the average unit costs of treating eye conditions were 10% higher (lower) than the figures in table 2.

The set of assumptions used by Shamanna et al[17] on lost productivity among people with diabetes who are blind was considered, as explained above, as an alternative to the Eckert et al[24] assumptions.

Alternative assumptions on the derivation of estimated loss of QALYs from blindness and MSVI were examined as presented in the online supplemental material. In this sensitivity analysis, equating blindness with 'I am unable to see' and MSVI with 'I have some problems seeing' but sharing the 'I have severe problems seeing' category

equally between those who are blind and those who have MSVI are continued. A second definition of the counterfactual was also employed in which the 'I have slight problems seeing' category was added to the 'I have no problems seeing' group.

### Patient and public involvement

Patients and the public were not involved in this study.

## RESULTS

### Prevalence of blindness among people with diabetes

This study considered people residing in India aged 40 years and over who have diabetes. Based on age-stratified prevalence data from the GBD for 2019 and age-stratified population data from PopulationPyramid.net for 2020, it is estimated that there were 55.9 million people aged 40 years and over and almost 41.2 million people aged 50 years and over in India with diabetes in 2020. As presented in table 1 above, using GBD data, it is estimated that of these 41.2 million people aged 40 years and over, there are some 1.34 million blind people with diabetes and 9.10 million people with MSVI and diabetes. The breakdown of these estimates by eye condition causing blindness or MSVI is also shown in table 1.

### Cost of screening and follow-up treatment

It is assumed that annual retinal screening will be offered to all 55.9 million people with diabetes aged 40 years and over. The annual cost of screening this group would be 22.3 billion INR assuming the cost of retinal screening per person is 400 INR on the basis of the Kerala pilot.

On the basis of the Kerala pilot screening programme in which 31% of those screened were referred for an eye examination, an estimated 17.3 million people of the 55.9 million people screened would be referred to hospital services for an eye examination. If all of these attended hospital for an eye examination at a cost of 1150 INR per individual, the cost of eye examinations for this group would be 19.9 billion INR. The estimated total cost of annual screening followed by eye examination where required would therefore be around 42.3 billion INR. These data are presented in table 3 below.

The cost of treating those found to have DR, cataract or glaucoma and the cost of glasses for those with refractive disorders, using the prevalence rates and unit costs of treatment shown above, are also considered. Treatment for age-related macular degeneration or for people with other eye conditions is not considered. Only people aged 50 years and over are considered since the GBD does not present data for people aged 40–49 years. Since the prevalence of blindness and MSVI in this group is likely to be low, the omission of this group is unlikely to have a large impact on the numbers of people treated.

It seems improbable that all those with prevalent eye conditions could be treated in 1 year and it is likely to require many years for the backlog to be cleared. On the basis that 10% could be treated each year, the cost would

**Table 3** Cost of screening for sight problems among people with diabetes aged 40+ years

| Item | Annual Number (millions) | Cost (million INR) | | |
|---|---|---|---|---|
| | | Healthcare | Travel | Total cost |
| Screening | 55.87 | 22 350 | | 22 350 |
| Eye examination | 17.32 | 3460 | 16 450 | 19 920 |
| Total | | 25 810 | 16 450 | 42 270 |

All people with diabetes aged 40+ years are assumed to be screened; 31% of those screened are assumed to need eye examination, as per Kerala pilot.
INR, Indian rupee.

be 1.43 billion INR per year, including 110 million INR for patient and attendant travel and loss of income. On the basis that 20% of the backlog could be treated each year, the cost would be 2.87 billion INR per year, including 220 million INR for patient and attendant travel and loss of income (table 4). Almost 73% of this cost is related to cataract surgery.

### Economic activity lost
The value of economic activity lost due to blindness and MSVI among people with diabetes who are blind or have MSVI is also estimated, following the method described in the Costing economic losses section.

The estimates are based on the following:
a. Economic activity lost due to blindness using Eckert et al[24] assumptions.
b. Economic activity lost due to blindness using Shamanna et al[17] assumptions.
c. Economic activity lost due to MSVI using Eckert et al assumptions.

It is assumed that output is lost only in the case of people aged 50–69 years and those over 70 years would

**Table 4** Cost of treating sight problems among diabetics aged 50+ years in India, if 20% of those needing treatment are treated each year

| Intervention | Annual Number (thousands) | Cost (million INR) | | |
|---|---|---|---|---|
| | | Healthcare | Travel | Total cost |
| Cataract surgery | 151.60 | 2270 | 140 | 2420 |
| STDR treatment | 25.27 | 300 | 70 | 380 |
| Glaucoma surgery | 3.69 | 20 | 4 | 20 |
| Glaucoma eye drops | 14.75 | 7 | — | 7 |
| Glasses | 42.84 | 40 | — | 40 |
| Total | 238.15 | 2650 | 220 | 2870 |

INR, Indian rupee; STDR, sight-threatening diabetic retinopathy.

not work even if they had better vision. Data are used on the numbers of people who experience blindness and MSVI caused by DR, by age and from South-East Asia,[26] since we could not find data by age for all people with diabetes and blindness or MSVI in India. In these data, 29.5% blind people aged 50+ years with DR are over 70 years and 32.0% of those with MSVI aged 50+ years with DR are over 70 years. Therefore, economic activity relating to 941 000 blind people and 6.186 billion people with MSVI is considered. Average GNI per capita as an indicator of the value of lost production is used. GNI per capita was 147 524 INR in 2019.[25]

The economic loss due to blindness is estimated at 153 billion INR using Eckert et al assumptions or 146 billion INR under Shamanna et al assumptions, and the economic loss due to MSVI is estimated at 319 billion INR under the Eckert et al assumptions (table 5).

### Loss of QALYs
The QoL of a blind person is estimated to be 0.395 lower than that of a person with no sight problem after standardising by age (since blind people are on average older than those without sight problems). Blindness thus means that for each year a blind person lives, they enjoy 0.395 fewer QALYs than a person without sight problems. The derivation of this estimate is set out in the online supplemental material. For 1.34 million blind people with diabetes aged 40 years and over, 0.53 million QALYs are lost per year due to blindness. Similarly, it is estimated that the QoL of a person with MSVI is 0.256 lower than that of a person with no sight problem, after standardising for age. For 9.10 million people with MSVI and diabetes aged 40+ years, 2.33 million QALYs are lost per year due to MSVI.

### Sensitivity analyses
If the average costs per person of screening for DR and eye examination where required were 10% higher (lower) than the 400 INR and 1150 INR, respectively, estimated for Kerala, the overall cost of screening and eye examinations would be 46.5 billion INR (38.0 billion INR) instead of 42.3 billion INR.

If the average unit costs of treating eye conditions were 10% higher (lower) than the figures in table 5, the overall cost of treatment would be 3.16 billion INR (2.58 billion INR) instead of 2.87 billion INR.

The alternative set of assumptions used by Shamanna et al on lost productivity among people with diabetes who are blind was also considered, as explained above. Under those assumptions, the estimated loss due to blindness would be 146 billion INR, compared with 153 billion INR using Eckert et al assumptions. However, if people with MSVI could produce 85% of the output of those with no visual impairment, as opposed to 70% assumed by Eckert et al, the loss of output due to blindness and MSVI would be 335 billion INR rather than 472 billion INR; but if 5% of those with MSVI could not work at all, the loss of output would be 536 billion INR.

**Table 5** Economic loss as a consequence of blindness or MSVI among the population of India aged 50+ years with diabetes (million INR)

| | Loss due to blindness (Eckert et al[24] assumptions) | Loss due to blindness (Shamanna et al[17] assumptions) | Loss due to MSVI (Eckert et al[24] assumptions) |
|---|---|---|---|
| Lost productivity of people with sight loss aged 50–69 unable to work | 138 900 | 111 100 | 0 |
| Lost productivity of people with sight loss aged 50–69 able to work partially | 0 | 20 800 | 273 800 |
| Lost productivity of family carers for all people with blindness or MSVI | 13 900 | 13 900 | 45 600 |
| Total productivity lost | 152 800 | 145 800 | 319 400 |

INR, Indian rupee; MSVI, moderate to severe visual impairment.

Additionally, alternative assumptions on the derivation of estimated loss of QALYs from blindness and MSVI were examined as discussed in the online supplemental material. Under those variants, the loss was in the range of 2.5–3.0 million QALYs per year.

## DISCUSSION

In this study, estimates of several types of cost that occur as a result of blindness and MSVI among people aged 40 years and over with diabetes in India are provided. If 100% of eligible people attended, annual screening followed by eye examination where required would cost around 42.3 billion INR per year, and treating sight problems among people with diabetes would cost 2.9 billion INR per year, if 20% of those needing treatment are treated annually. The estimated lost economic activity per year is 472 billion INR, and loss of QoL is estimated at around 2.86 million QALYs per year. The estimate of lost production, however, is highly sensitive to the proportion of people with MSVI able to work and how their output compares with that of a person with no visual impairment.

### Previous studies

There have been previous studies looking at the economic burden of diabetes, blindness and DR in countries across the world but none to our knowledge that have looked at this issue specifically in India. Also, those studies identified tend to focus on specific aspects of cost, which makes comparisons with this study difficult.

Toth et al[12] found that the direct cost of treating DR in Hungary was US$145.8 million in 2016 and will increase to US$169.0 million by 2045 but did not estimate lost economic activity. Work in Singapore[13] shows that, on average, people with DR use medical care costing almost four times as much as a similar group of patients with diabetes without DR.

The total national cost of diagnosed type 2 diabetes in Iran was estimated at US$3.78 billion in 2009,[14] consisting of US$2.04 billion direct (medical and non-medical) costs and US$1.73 billion indirect costs. Ophthalmic complications made up 14% of this figure

and permanent disability, of which blindness would be a major proportion, was responsible for US$452.4 million of the indirect costs. Research from Germany[15] estimated that the total societal cost of DR was €3.51 billion for the year 2002.

Shamanna et al[17] estimated that the burden of blindness—in terms of lost output—in India for the year 1997 was 159 billion INR (US$4.4 billion), which equates to 665 billion INR at 2020 prices.[27] Shamanna et al, however, while not covering MSVI, considered all blind people of all ages and not just blindness among people aged 40+ years with diabetes.

There have been several studies of related topics in the USA. Schmier et al[28] found that Medicare payments were on average $182 higher per year among claimants with diabetes with non-proliferative retinopathy than those without ($10 613 vs $9981), and $3825 higher among patients with proliferative retinopathy ($13 806).

Lee et al[16] found that among a sample of US employees in 2004, direct and indirect annual costs of treating employees with DR ($18 218 and $3548) were significantly higher than those without DR (direct=$11 898; indirect=$2374).

It was estimated that in 2004,[11] the direct medical costs of treating DR among Americans aged 40–64 years were approximately $300 million, although more recent work suggests that this is a very small proportion of the burden caused by diabetes as a whole—the total estimated economic burden of diagnosed diabetes in the USA in 2017[29] was $327 billion, which consisted of $237 billion in direct medical costs and $90 billion in reduced productivity.

A paper relevant to this work by Orji et al[30] analysed the cost of treating people with DR at a tertiary eye care centre in South India. They considered direct costs of the consultation, investigations, intravitreal injections and vitreoretinal surgeries. Furthermore, they included indirect costs of transportation and boarding-lodging costs (estimates from a nearby hotel). Their median cost was 8214 INR but includes patients who received a wide spectrum of treatments.

## Strengths and limitations

The major strength of this paper is that it is, to the best of our knowledge, the first to estimate the costs of DR screening and treatment, the value of economic activity lost and the value of QALYs lost due to blindness and MSVI among people with diabetes in India.

This study has been able to use data on the costs of treating patients with DR from a recent new pilot scheme that has been operating in Kerala that uses accessible technology to screen patients with diabetes and refer them where necessary for follow-up investigation and treatment in hospital. While detailed recent data from Kerala are used, it cannot be certain that the costs of screening would be similar across other states in India.

The estimate of the value of lost productivity uses recent data on the prevalence of blindness among people with diabetes and of average incomes in India. A significant limitation however is that, in the absence of evidence on the proportion of blind people with diabetes and of people with MSVI and diabetes who are unable to work, assumptions have been required, the same assumptions as Eckert *et al* employed. An alternative considered was to use data from the 2011 Census instead of those assumptions, but there was concern from experts about the plausibility of Census data, as the Census uses different definitions of blindness and sight loss. This is an area that deserves further exploration.

The estimate is related to the current position. If screening was introduced for all people aged 40+ years with diabetes, there would in due course be fewer blind people and people with MSVI with diabetes, and both the QALYs lost and the economic activity lost would be lower. On the other hand, the number of people in India with diabetes is expected to grow significantly in the next decades,[31] leaving more people at risk of DR.

There is a potential issue with overlap between estimated QALY loss and economic activity loss such that summing both elements would be controversial and does not seem appropriate. Caution is required when adding the estimated cost of screening and treatment to either the estimated QALY loss or economic activity loss, since the costs are related to different groups of people with diabetes. The costs of screening are related to people with diabetes whether or not they have sight problems. The costs of treatment are related mainly to people who have MSVI and are at risk of blindness. The costs of lost productivity and lost QALYs are related to people who are blind or have MSVI. In practice, the treatment costs that would be incurred annually would be the costs of treating those newly diagnosed following screening, which would for at least several years include part of a backlog of people who have had deteriorating sight which could have been detected earlier if there had been screening. It is assumed that in the early years, some 10%–20% of those needing treatment would be treated each year.

## Implications

The most important findings of this study are that the annual cost to the Indian economy arising from lost productivity due to blindness and MSVI among people with diabetes is substantial, and that the cost of annual screening of people with diabetes for eye conditions and treating those with sight-threatening DR, cataract or glaucoma would also be substantial. Over time, however, the costs of treatment are likely to fall, when the backlog of people with diabetes with untreated eye conditions is cleared. This is before considering the impact of demographic change and inflation. Also, over time, the costs incurred through lost productivity will also decline, as the proportion of people with diabetes who become blind falls due to regular screening and timely treatment of those at risk of losing their sight. A policy to conduct annual screening followed by treatment where required would therefore yield a net benefit to the Indian economy as well as a benefit to people with diabetes at risk of blindness and their families.

**Acknowledgements** The authors would like to thank the experts who contributed information relating to the cost of ophthalmology treatment in India.

**Collaborators** ORNATE India Project Group: Dr Bipin Gopal, Directorate of Health Services, Thiruvananthapuram, Kerala; Mr Rajeev Sadanandan, Health Systems Transformation Platform, New Delhi; Dr Vasudeva Iyer Sahasranamam, Regional Institute of Ophthalmology, Thiruvananthapuram, Kerala; Dr Simon George, Regional Institute of Ophthalmology, Thiruvananthapuram, Kerala; Professor Gopalakrishnan Netuveli, Institute for Connected Communities, University of East London, London; Dr Ramachandran Rajalakshmi, Department of Diabetology, Ophthalmology and Epidemiology, Madras Diabetes Research Foundation and Dr Mohan's Diabetes Specialities Centre, Chennai; Dr Deepa Mohan, Department of Diabetology, Ophthalmology and Epidemiology, Madras Diabetes Research Foundation and Dr Mohan's Diabetes Specialities Centre, Chennai; Dr Viswanathan Mohan, Department of Diabetology, Ophthalmology and Epidemiology, Madras Diabetes Research Foundation and Dr Mohan's Diabetes Specialities Centre, Chennai; Dr Taraprasad Das, Anant Bajaj Retina Institute-Srimati Kanuri Santhamma Centre for Vitreoretinal Diseases, Hyderabad Eye Research Foundation, LV Prasad Eye Institute, Hyderabad; Dr Dolores Conroy, Vision Sciences, UCL Institute of Ophthalmology, London, UK; Miss Lakshmi Premnazir, Directorate of Health Services, Thiruvananthapuram Kerala; Ms Jyotsna Srinath, Institute for Connected Communities, University of East London, London, UK; Dr Pramod Bhende, Sankara Nethralaya, Chennai, Tamil Nadu; Ms Janani Surya, Sankara Nethralaya, Chennai, Tamil Nadu; Mrs Radha Ramakrishnan, Vision Sciences, UCL, London; Dr Rupak Roy, Sankara Nethralaya, Kolkata; Dr Supita Das, Sankara Nethralaya, Kolkata; Dr George Manayath, Aravind Eye Hospital, Coimbatore, Tamil Nadu; Dr Vignesh T Prabhakaran, Aravind Eye Hospital, Madurai, Tamil Nadu; Dr Giridhar Anantharaman, Giridhar Eye Institute, Cochin, Kerala; Dr Mahesh Gopalakrishnan, Giridhar Eye Institute, Cochin, Kerala; Dr Sundaram Natarajan, Aditya Jyot Hospital, Mumbai, Maharashtra; Dr Radhika Krishnan, Aditya Jyot Hospital, Mumbai, Maharashtra; Dr Sheena Liz Mani, Dr Tony Fernandez Eye Hospital, Aluva, Kerala; Dr Manisha Agarwal, Dr Shroff's Charity Eye Hospital, New Delhi; Dr Padmaja Kumari Rani, LV Prasad Eye Institute, Hyderabad, Telangana; Dr Umesh Behera, LV Prasad Eye Institute, Bhubaneshwar, Odisha; Dr Harsha Bhattacharjee, Sri Sankaradeva Nethralaya, Guwahati, Assam; Dr Manabjyoti Barman, Sri Sankaradeva Nethralaya, Guwahati, Assam; Dr Gajendra Chawla, Vision Academy–The Socio Medical Society, Bhopal, Madhya Pradesh; Dr Alok Sen, Sadguru Netra Chikitsalaya, Chitrakoot, Madhya Pradesh; Dr Moneesh Saxena, Aurobindo Nethralaya, Raipur, Chhattisgarh; Dr Asim K Sil, Netra Niramay Niketan, Haldia, West Bengal; Dr Subhratanu Chakabarty, Netra Niramay Niketan, Haldia, West Bengal; Dr Thomas Cherian, Little Flower Hospital and Research Centre, Angamaly, Kerala; Dr Reesha Jitesh, Little Flower Hospital and Research Centre, Angamaly, Kerala; Dr Rushikesh Naigaonkar, Netra Niramay Niketan, Haldia, West Bengal; Dr Abishek Desai, Netra Niramay Niketan, Haldia, West Bengal; Dr Sucheta Kulkarni, HV Desai Hospital, Pune, Maharashtra.

**Contributors** SR acts as guarantor, developed the original concept, acquired data, performed analysis, drafted the paper and approved the submitted version. RA

contributed to elements of the original concept, performed analysis, contributed to the drafting of the paper and approved the submitted version. RR contributed to acquiring cost data and approved the submitted version. SS advised on the concept, provided expert guidance throughout, acquired data and approved the submitted version. RW contributed to the original concept, acquired data, performed analysis, drafted parts of the paper and approved the submitted version.

**Funding** This work is part of the ORNATE India Project funded by the Global Challenges Research Fund (GCRF) UK Research and Innovation (UKRI) (MR/P207881/1). The research is supported by the NIHR Biomedical Research Centre at Moorfields Eye Hospital NHS Foundation Trust and UCL Institute of Ophthalmology.

**Competing interests** None declared.

**Patient and public involvement** Patients and/or the public were not involved in the design, or conduct, or reporting, or dissemination plans of this research.

**Patient consent for publication** Not required.

**Ethics approval** This study does not involve human or animal participants; thus, ethical approval is not required.

**Provenance and peer review** Not commissioned; externally peer reviewed.

**Data availability statement** Data are available upon reasonable request. The data used in this study are available upon request.

**ORCID iDs**
Stuart Redding http://orcid.org/0000-0001-5974-4395
Robert Anderson http://orcid.org/0000-0002-8580-9663
Rajiv Raman http://orcid.org/0000-0001-5842-0233
Sobha Sivaprasad http://orcid.org/0000-0001-8952-0659
Raphael Wittenberg http://orcid.org/0000-0003-3096-2721

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
