## [Reviewer comments · BMJ Open]

ARTICLE DETAILS

TITLE (PROVISIONAL)	ESTIMATING THE COSTS OF BLINDNESS AND MODERATE TO SEVERE VISUAL IMPAIRMENT AMONGST PEOPLE WITH DIABETES IN INDIA
AUTHORS	Redding, Stuart; Anderson, Robert; Raman, Rajiv; Sivaprasad, Sobha; Wittenberg, Raphael

VERSION 1 – REVIEW

REVIEWER	Marques, Ana Patricia London School of Hygiene and Tropical Medicine International Centre for Eye Health
REVIEW RETURNED	25-Nov-2022

GENERAL COMMENTS	The manuscript reports results from a cost of illness study of people with diabetes and vision problems aged 40 and above in India. This is an interesting study that combines data from a pilot screening study in India, a community screening programme in India, published literature and expert advice. Overall, I found the manuscript to be well written and well-structured. I have a few suggestions mainly to help clarify methodological options and assumptions and to expand results reporting around the sensitivity analysis. Main text Introduction: - Minor issues - The introduction give us enough information to understand the topic, provides relevant information about the context and available literature and presents the study objectives and its relevance for health policy. In my opinion, authors not always provide sufficient references to support their statements. Here are a few examples: Introduction – Manuscript page 3 -Lines 30 to 39 – This is the first paragraph; it presents WHO Blindness definition and Blindness impact and consequences in terms of costs and potential productivity losses but there isn't ANY reference to support this opening statement. I suggest that the authors consider adding references to this statement and revise the wording about the impact of blindness in terms of economic productivity losses. There are people with blindness and moderate to severe vision impairment that have a full-time job and contribute to economic productivity of their countries although this is not the case for all people with vision impairment. So, I understand and agree with what authors are saying I just think it is important not to stay and assume that
---

	people with VI are not contributing at all to the society and economy. Manuscript page 4 -Lines 22 to 42 – No references on cost of illness definition. Methods: Manuscript Page 8 line 9 and 10 – “ in the table imply rates of 3.07 People with diabetes who have MSVI.” - This statement is not clear to me. In table 1 you have a prevalence rate per 1000 population for DR and blindness of 0.472 and 1.21 for DR and MSVI. Please clarify what do you want to say here. Manuscript Page 8 line 16 and 17 - Check the £ symbol - is this correct? I read it as million people not million £. I also think that it is not very informative just to state that the higher estimate of blind people with diabetes was used to avoid under-estimation. It is not correct to produce either under or over estimates. Since there is a considerable variation between these two figures - 1.225 million or 1.335 million – I recommend that the prevalence of blindness is included in a sensitivity analysis. Manuscript Page 8 line 48 to 52 - these calculations should be described and presented in supplementary data they are a key element of the study- it should also be explained why cost data was driven by Kerala when Reference 8 reports a community screening conducted in 20 regions in India Manuscript Page 9 – Table 2 Please provide more details about the use of expert advice namely how, who, how many experts were used to obtain these estimates. Please explain why eye drops costs weren't also included in the expert panel exercise. Manuscript Page 10 line 40 to 52 – I think it would be useful to explain: 1) why did the authors felt the need to use their assumptions; 2) what is the rationale to assume these numbers and not others. Manuscript Page 11 line 54 and 55 – Please quantify some. Results Manuscript Page 13 line 60 – Is this correct? Table 5 shows the cost of treating sight problems amongst If 20% .. are treated. Manuscript Page 14 line 3 and 4 – Is this correct? Table 4 shows Cost of screening for sight problems amongst people with diabetes aged 40+ Manuscript Page 15 lines 40 to 60. Loss of quality adjusted life years (QALYS). Is is unclear how authors have reached to a difference of 0.395 (for blind people) and 0.256 (for people with MSVI) when compared to people without sight problems. I recommend that authors explain in a more detailed manner how these results were obtained. It is not enough to mention at the methods section that individual data was obtain from the SMART Study (using EQ5D5L and Manuscript Page 16 lines 12 to 20 – Sensitivity analysis results show be reported more carefully. The same scenarios presented in Table 6 should be reproduced for different vision problems
--	---

	grouping and the estimate around prevalence - should also be taken into consideration.
--	--

VERSION 1 – AUTHOR RESPONSE

Reviewer: 1

Ms. Ana Patricia Marques, London School of Hygiene and Tropical Medicine International Centre for Eye Health

Comments to the Author:

BMJ Open, Title: "Estimating the costs of blindness and moderate to severe visual impairment amongst people with diabetes in India"

The manuscript reports results from a cost of illness study of people with diabetes and vision problems aged 40 and above in India. This is an interesting study that combines data from a pilot screening study in India, a community screening programme in India, published literature and expert advice.

Overall, I found the manuscript to be well written and well-structured. I have a few suggestions mainly to help clarify methodological options and assumptions and to expand results reporting around the sensitivity analysis.

RESPONSE: We thank the reviewer for these comments and for her helpful advice on how to improve the paper. We set out below how we have addressed her comments.

Main text

Introduction: - Minor issues -

The introduction give us enough information to understand the topic, provides relevant information about the context and available literature and presents the study objectives and its relevance for health policy. In my opinion, authors not always provide sufficient references to support their statements. Here are a few examples:

Introduction –

Manuscript page 3 -Lines 30 to 39 – This is the first paragraph; it presents WHO Blindness definition and Blindness impact and consequences in terms of costs and potential productivity losses but there isn't ANY reference to support this opening statement.

I suggest that the authors consider adding references to this statement and revise the wording about the impact of blindness in terms of economic productivity losses. There are people with blindness and moderate to severe vision impairment that have a full-time job and contribute to economic productivity of their countries although this is not the case for all people with vision impairment. So, I understand and agree with what authors are saying I just think it is important not to stay and assume that people with VI are not contributing at all to the society and economy.

RESPONSE: We have added a reference to show where the WHO definition of blindness comes from and revised the text to clarify that people with VI often contribute to the economy, specifically:

Blindness, defined by the World Health Organisation as Snellen visual acuity of 3/60 or worse in the better eye¹, is a debilitating condition which, in addition to the disabling consequences for the person affected, generates substantial care and treatment costs for society. While some people who suffer from vision loss are able to work, their productivity is likely to be less than that of a person with better vision. and some may drop out of the labour force entirely. Therefore blindness and sight loss reduces the potential output of the economy by decreasing the output of both those who are directly affected and those who have to stop or reduce work in order to provide unpaid care for people who are blind.

Manuscript page 4 -Lines 22 to 42 – No references on cost of illness definition.

RESPONSE: We have added a reference that provides “a summary of the approaches utilised and their limitations”. The definition is not a direct quote but is in our words.

Methods:

Manuscript Page 8 line 9 and 10 – “ in the table imply rates of 3.07 People with diabetes who have MSVI.” - This statement is not clear to me. In table 1 you have a prevalence rate per 1000 population for DR and blindness of 0.472 and 1.21 for DR and MSVI. Please clarify what do you want to say here.

RESPONSE: The figures of 0.472 and 1.21 are calculated by dividing the number of people with DR by the total population aged 50 and over. The figures 3.07 and 7.86 are calculated by dividing the number of people with DR by the number of people with diabetes aged 50 and over.

We have amended this paragraph and changed the column titles on table 1 to clarify these figures.

Manuscript Page 8 line 16 and 17 - Check the £ symbol - is this correct? I read it as million people not million £.

RESPONSE: This should be million people not million £ and we have corrected this in the text.

I also think that it is not very informative just to state that the higher estimate of blind people with diabetes was used to avoid under-estimation. It is not correct to produce either under or over estimates. Since there is a considerable variation between these two figures - 1.225 million or 1.335 million – I recommend that the prevalence of blindness is included in a sensitivity analysis.

RESPONSE: On reviewing the way we derived prevalence numbers from prevalence rates in the Global Burden of Disease paper, we realise that the derivation of the lower number was a mistake on our part and that we should use only the higher number.

Manuscript Page 8 line 48 to 52 - these calculations should be described and presented in supplementary data – they are a key element of the study- it should also be explained why cost data was driven by Kerala when Reference 8 reports a community screening conducted in 20 regions in India

RESPONSE: We have added material summarising the basis for our estimate of the costs of screening for diabetic retinopathy in the Kerala pilot screening programme. We are submitting to a journal another paper which presents and discusses the costs and outcomes of the Kerala pilot. We are in the process of submitting this to a pre-print server and will be able to include reference to that version of the paper when it comes online. We cannot use data from the SMART study conducted in 20 areas since the screening in that study was for diabetes and its sequelae and not just for DR.

Manuscript Page 9 – Table 2 Please provide more details about the use of expert advice, namely how, who, how many experts were used to obtain these estimates. Please explain why eye drops costs weren't also included in the expert panel exercise.

RESPONSE: The expert advice on unit costs was provided by a group comprising the lead researchers from each of the 20 sites in the SMART study, except that for eye drops the advice was provided by the PI in consultation with some of the SMART study leads. We have amended the text to explain who provided the expert advice.

Manuscript Page 10 line 40 to 52 – I think it would be useful to explain: 1) why did the authors feel the need to use their assumptions; 2) what is the rationale to assume these numbers and not others.

RESPONSE: We have removed the estimate we calculated using our assumptions of the loss of productivity associated with MSVI. The Shamanna paper only considered loss associated with blindness, not MSVI. We have now reordered the estimates so that Eckert is first since this paper considers both blindness and MSVI.

Manuscript Page 11 line 54 and 55 – Please quantify some.

RESPONSE:

We have amended the text to make clear that we assume that all those in vision categories 4 and 5 are blind and all those in vision category 3 have MSVI. In the sensitivity analysis section we investigate an alternative assumption in which half of those in vision category 4 and all those in vision category 5 are blind and all those in vision category 3 and half those in vision category 4 have MSVI. We then assume in this sensitivity analysis that the EQ5D HRQoL is the same for those in vision category 4 who are blind and those in vision category 4 who have MSVI, since it is not obvious what alternative assumption would be better.

Results

Manuscript Page 13 line 60 – Is this correct? Table 5 shows the cost of treating sight problems amongst If 20% .. are treated.

RESPONSE: This was a mistake on our part. We do not present a table with the values if 10% of those needing treatment are treated each year, so we have removed this reference.

Manuscript Page 14 line 3 and 4 – Is this correct? Table 4 shows Cost of screening for sight problems amongst people with diabetes aged 40+

RESPONSE: As discussed above, this should say Table 5.

Manuscript Page 15 lines 40 to 60. Loss of quality adjusted life years (QALYS). It is unclear how authors have reached to a difference of 0.395 (for blind people) and 0.256 (for people with MSVI) when compared to people without sight problems. I recommend that authors explain in a more detailed manner how these results were obtained. It is not enough to mention at the methods section that individual data was obtained from the SMART Study (using EQ5D5L)

RESPONSE: We have set out a detailed explanation in a supplementary note to accompany the paper and added a brief explanation to the main text.

Manuscript Page 16 lines 12 to 20 – Sensitivity analysis results should be reported more carefully. The same scenarios presented in Table 6 should be reproduced for different vision problems grouping and the estimate around prevalence - should also be taken into consideration.

RESPONSE: We have expanded the sensitivity analysis accordingly.

In addition to these changes we have added a list of collaborators to the manuscript which we omitted from the original submission. These contributors made an important contribution by collecting data without which we could not have produced this paper.

VERSION 2 – REVIEW

REVIEWER	Marques, Ana Patricia London School of Hygiene and Tropical Medicine International Centre for Eye Health
REVIEW RETURNED	27-Feb-2023

GENERAL COMMENTS	Thank you for addressing my previous comments/suggestions. I have only a few more minor comments and suggestions. My main suggestion is to consider adding an item at the methods section describing all the sensitivity analysis that was done. There is a description regarding lost productivity estimates but, apart from that, there is no information about the sensitivity analysis that was performed for screening costs, treatment costs and alternative options for deriving QALYs loss. Page 4 – line 99 to 106
---

	It is stated that “ Our cost of illness study estimates and lost productivity by those who are blind”. Since, for lost productivity estimates it has been considered people with blindness and people with MSVI, as mention in a sentence bellow (lines 104 to 106) as well every else in the manuscript, I suggest to consider editing the first sentence by explicitly mentioning these subgroup of patients. Page 10 – 2.2.4 – Calculating QALY losses Please consider to add a brief description of how did you to estimate Qalys differences between a blind person and a person with no sight problem or at least mention at the methods section that further details are given in supplementary data. I think it might also be worth to explain how these differences were standardized by age (as mentioned solely on the results section page 16 lines 387 to 389). Page 12 – lines 321 to 323 and Table 3. I might be missing something here but I could not find the information about the number of people with MSVI and diabetes in Table 3, although it was quite easy to locate both number of people with diabetes in India and number of people with diabetes and all cause of blindness. Page 13 - 3.3 Economic activity lost Lines 363 to 364 – I suggest including people with MSVI in this sentence too. Page 16 – lines 395 to 396 I don’t see the relevance of converting Qalys loss into monetary terms. This was not describe in the methods section nor included at the discussion section.
--	--

VERSION 2 – AUTHOR RESPONSE

Reviewer: 1

Ms. Ana Patricia Marques, London School of Hygiene and Tropical Medicine International Centre for Eye Health

Comments to the Author:

Thank you for addressing my previous comments/suggestions. I have only a few more minor comments and suggestions.

My main suggestion is to consider adding an item at the methods section describing all the sensitivity analysis that was done. There is a description regarding lost productivity estimates but, a part from that, there is no information about the sensitivity analysis that was performed for screening costs, treatment costs and alternative options for deriving QALYs loss.

RESPONSE: We have added a new section 2.2.5 to the methods in which we outline the sensitivity analyses performed in this paper.

Page 4 – line 99 to 106

It is stated that “Our cost of illness study estimates and lost productivity by those who are blind”. Since, for lost productivity estimates it has been considered people with blindness and people with MSVI, as mention in a sentence below (lines 104 to 106) as well every else in the manuscript, I suggest to consider editing the first sentence by explicitly mentioning these subgroup of patients.

RESPONSE: We have added to the sentence mentioned so that it now says:

“Our cost of illness study estimates the total costs of screening people aged 40 and over with diabetes, healthcare for those found to have eye conditions and lost productivity by those who are blind or experience MSVI and by family and friends providing unpaid care for them.”

Page 10 – 2.2.4 – Calculating QALY losses

Please consider to add a brief description of how did you to estimate QALYs differences between a blind person and a person with no sight problem or at least mention at the methods section that further details are given in supplementary data.

I think it might also be worth to explain how these differences were standardized by age (as mentioned solely on the results section page 16 lines 387 to 389).

RESPONSE: We have edited the final sentence of this section so that it now reads:

“To estimate the loss of QALYs due to blindness and MSVI we treat the response category “I have no problems seeing” is treated as the counterfactual against which to estimate the loss. The Chinese value set [26] was applied to the EQ5D data for each person to get a quality of life (QoL) value for each person. Since the aim is to derive the average reduced quality of life due to poor sight in poor sight it is necessary to establish the average QoL those with poor sight would have had if they had good sight. Since their average age is greater than those with good sight, and QoL level falls with age, the age distribution of those with poor sight is applied to the age specific QoL values in those with good sight to derive the appropriate comparator – the average QALY if those with poor sight had good sight. Further details of the methods employed in this analysis are presented in the supplementary material.”

Page 12 – lines 321 to 323 and Table 3.

I might be missing something here but I could not find the information about the number of people with MSVI and diabetes in Table 3, although it was quite easy to locate both number of people with diabetes in India and number of people with diabetes and all cause of blindness.

RESPONSE: We have added a line to the table with this information.

Page 13 - 3.3 Economic activity lost

Lines 363 to 364 – I suggest including people with MSVI in this sentence too.

RESPONSE: We have added to this sentence so that it now reads “We also estimate the value of economic activity lost due to blindness (or MSVI) amongst people with diabetes who are blind (or have MSVI).

Page 16 – lines 395 to 396

I don't see the relevance of converting Qaly loss into monetary terms. This was not described in the methods section nor included at the discussion section.

RESPONSE: We have deleted this sentence.